# The Effect of Ketoconazole and Quinestrol Combination on Reproductive Physiology in Male Mice

**DOI:** 10.3390/ani14223240

**Published:** 2024-11-12

**Authors:** Yu Ji, Yujie Wang, Yuhang Liu, Yutong Liu, Jiao Qin, Daohuan Yuan, Quansheng Liu

**Affiliations:** Guangdong Key Laboratory of Animal Conservation and Resource Utilization, Guangdong Public Laboratory of Wild Animal Conservation and Utilization, Institute of Zoology, Guangdong Academy of Sciences, Guangzhou 050000, China; 18048292741@163.com (Y.J.); wangyj.prc@foxmail.com (Y.W.); 13548055660@163.com (Y.L.); 13347318892@163.com (Y.L.); qjohy@163.com (J.Q.); yuandaoh@126.com (D.Y.)

**Keywords:** fertility control, ketoconazole, quinestrol, CYP3A4

## Abstract

This study explores a new way to control fertility in male mice by combining two substances: quinestrol, a synthetic hormone, and ketoconazole, a medication usually used to treat fungal infections. The researchers wanted to see if adding ketoconazole could make quinestrol more effective, so less of the hormone would be needed. They tested different combinations of these two substances on male mice and checked their impact on fertility-related factors like sperm count and reproductive organ size. The most promising results came from a low dose of quinestrol combined with ketoconazole, which significantly reduced sperm count and lowered testosterone levels after 10 days. Even after 30 days, this combination continued to exhibit some suppressive effects on reproductive capacity. The study also found that ketoconazole helped to slow down the body’s metabolism of quinestrol, making it work more effectively. This study suggests that using ketoconazole alongside quinestrol could be a safer and more efficient way to control fertility, potentially offering a new approach for managing populations of certain animal species.

## 1. Introduction

Rodents, characterized by their adaptability and high reproductive rates, are prone to rapid and widespread proliferation under favorable conditions [1]. As a result, rodent infestations cause substantial economic damage to agriculture and the environment [2,3] Furthermore, rodents transmit various diseases to humans and livestock, representing a global health threat [4]. Although chemical extermination remains the primary control method, it faces limitations, including short-term efficacy, environmental pollution, harm to non-target species, and the development of drug resistance [5]. Fertility control, as documented in numerous studies [6,7], is a promising, environmentally friendly, and humane alternative that aligns with Ecologically Based Rodent Management (EBRM) principles [8,9]. Fertility control regulates population density by reducing birth rates through surgical/chemical, endocrine disruption, and immunological methods [10]. Notably, endocrine disruption has demonstrated significant potential and cost-effectiveness [11], with advances in reproductive endocrinology presenting new opportunities for rodenticide development [12,13,14,15].

Quinestrol, a synthetic estrogen derived from ethinylestradiol, is known for its long-term contraceptive effects due to its slow release from adipose tissue after oral administration [16,17]. Its contraceptive efficacy has been validated across various rodent species [18,19,20,21,22]. In females, quinestrol disrupts estradiol secretion and function, inhibits the release of gonadotrophin-releasing hormone from the hypothalamus, and further inhibits follicular growth, causing uterine oedema and reducing fertility [23,24,25,26]. In males, quinestrol disrupts the hypothalamic–pituitary–testicular axis [27], downregulates FSHβ and LHβ mRNA expression [28], induces oxidative stress [29], and impairs reproductive organ development, causing testicular atrophy, ultimately leading to reduced sperm quality and inhibited reproductive hormone function [30]. Although quinestrol is highly effective in controlling rodent populations, current challenges include the need for high dosages and poor palatability. These issues not only elevate the cost of rodent control but also increase the risk of environmental residues [31,32]. The oral absorption rate of quinestrol is very high, approaching 100%, which means when quinestrol is administered orally, nearly all of the drug is absorbed and enters the bloodstream [33,34]. However, after entering systemic circulation, quinestrol undergoes first-pass metabolism in the liver, resulting in an oral bioavailability of approximately 38–48% [35,36]. The catabolism of estrogens is mediated by Cytochromes P450 (CYP450) enzymes [37,38]. The catabolism of quinestrol was predominantly carried out by CYP3A4 and CYP1A2 [20,39,40]. Inhibiting the activity of CYP3A4 is a possible way to reduce quinestrol dosage and enhance its effectiveness. CYP3A4 inhibitors can affect the activity of this enzyme, thereby affecting drug metabolism and therapeutic effects [41,42]. The coadministration of CYP3A4 inhibitors with other medications can enhance drug efficacy or reduce dosage while minimizing adverse effects [43,44,45]. Ketoconazole is a potent CYP3A4 inhibitor, and studies have shown that it has the potential to improve therapeutic efficacy by affecting the drug transporters and thus blocking the metabolism of the drug [46,47]. Whether the combination of ketoconazole and quinestrol produces a similar effect that reduces the degradation of quinestrol requires further investigation through additional experiments.

Quinestrol has demonstrated significant efficacy in impairing reproductive functions in various rodent models, establishing important dosing parameters for its application in fertility control. Sidhu et al. [19] illustrated that administering quinestrol at doses approaching 1 mg/kg and 5 mg/kg resulted in marked reproductive impairment in Bandicota bengalensis. Similarly, Zhao et al. [27] reported that a dosage of 0.34 mg/kg led to substantial reductions in sperm counts and reproductive organ weights in Lasiopodomys brandtii. Based on these findings, our study selected doses of 1 mg/kg and 5 mg/kg to facilitate a comprehensive evaluation of quinestrol’s reproductive effects. Previous research included studies that evaluated fertility in male rats 4 weeks post-administration [48,49], while others assessed fertility 30 days after treatment [50]. In contraceptive agent development, understanding the duration of action is crucial for evaluating practical applicability. This study aims to assess the short-term effects of quinestrol, focusing on 10- and 30-day observation periods to capture the initial physiological responses to treatment. While short-term data may not fully reflect long-term effects, they provide foundational insights into treatment mechanisms and efficacy. Additionally, understanding fertility control in male animals is essential for advancing reproductive health across various domains, including fields such as animal health management and human contraception [51,52]. Our study investigates whether the combination of ketoconazole, a potent CYP3A4 inhibitor, with quinestrol enhances reproductive suppression, allowing for reduced quinestrol dosages while maintaining effectiveness. Specifically, we explore the impact on reproductive function and duration when quinestrol was administered alone or in combination with varying doses of ketoconazole in male mice, aiming to optimize contraceptive efficacy while minimizing environmental impact and improving drug palatability.

## 2. Materials and Methods

### 2.1. Animals and Drugs

Kunming mice, bred from offspring obtained from the Guangdong Medical Laboratory Animal Center, were housed individually in opaque plastic cages (300 mm × 200 mm × 160 mm) with stainless steel mesh covers. Mice had unrestricted access to standard rodent chow and water. Bedding was changed weekly. The animal room was maintained at 25 ± 2 °C on a 12L/12D light cycle (lights on 8:00–20:00) with negative pressure ventilation and air exchange conducted four times daily for 30 min. All procedures were approved by the Ethics Committee of the Institute of Zoology, Guangdong Academy of Sciences Approval Code: GIZ20220821. Approval Date: 24 August 2022.

Quinestrol (99.9% purity, Beijing Zizhu Medicine Co., Ltd., Beijing, China) was dissolved in sunflower seed oil to prepare a 0.1 mg/mL solution. Ketoconazole (99.9% purity, Shandong Heze Juhe Industrial Co., Ltd., Heze, China) was similarly dissolved to prepare 0.4 mg/mL and 2.0 mg/mL solutions. Both solutions were individually sonicated at 40 kHz for 15 min until fully dissolved to ensure their proper dissolution and homogeneity.

### 2.2. Experimental Treatment

The 10th and 30th day post-administration are suitable time points for observing the effects on the mouse reproductive system, considering the time required for organ weight changes to occur. These time points capture different stages of reproductive system development and maturation, enabling assessment of infertility drug effects on various aspects, including testes and epididymis development, sperm production, seminal vesicle secretion, and long-term effects of the drug [15]. In this experiment, 104 healthy adult mice were randomly divided into 2 large groups, 1 group was used to assess the results at 10 days (n = 52) and the other was used to assess the results at 30 days (n = 52). To ensure homogeneity across groups, mice were allocated treatment and control groups based on their initial body weight and age. This approach minimizes variability, enabling more accurate comparisons of treatment effects. The weight range for each group was carefully controlled to prevent confounding factors related to size or developmental stage. The 52 mice in each group were randomly divided into 6 groups: control check (10 d:CK1; 30 d: CK2) (15), Q1 (7), Q1 + K0.4 (7), Q1 + K2 (8), Q5 (7), Q5 + K0.4 (8). CK (10 d:CK1; 30 d: CK2): sunflower seed oil, Q1: quinestrol 1.0 mg/kg+ sunflower seed oil, Q1 + K0.4: quinestrol 1.0 mg/kg + ketoconazole 0.4 mg/kg, Q1 + K2: quinestrol 1.0 mg/kg + ketoconazole 2.0 mg/kg, Q5: quinestrol 5.0 mg/kg + sunflower seed oil, and Q5 + K0.4: quinestrol 5.0 mg/kg + ketoconazole 0.4 mg/kg (Table 1). There was no significant difference in the weight of males between groups.

At first, male mice in treatment groups (Q1, Q1 + K0.4, Q1 + K2, Q5, and Q5 + K0.4), respectively, received gavage in a dosage of sunflower seed oil, ketoconazole 0.4 mg/kg, ketoconazole 2.0 mg/kg, sunflower seed oil, and ketoconazole 0.4 mg/kg. After 30 min, male mice in treatment groups (Q1, Q1 + K0.4, Q1 + K2, Q5, and Q5 + K0.4), respectively, received gavage in a dosage of quinestrol 1.0 mg/kg, quinestrol 1.0 mg/kg, quinestrol 1.0 mg/kg, quinestrol 5.0 mg/kg, and quinestrol 5.0 mg/kg. The male control (CK1 and CK2) received sunflower oil following the same protocol as the treatment groups. The treatment was administered for three consecutive days. The administration of ketoconazole and quinestrol was performed via oral gavage to ensure accurate and consistent delivery of the intended dosages. This method was selected to avoid issues related to the unpleasant odor of ketoconazole, which could reduce voluntary consumption. Trained personnel conducted the procedure, adjusting dosage volumes based on each animal’s body weight, ensuring that all received the full required dosage. The drugs were administered separately to prevent potential interactions that might affect the stability or efficacy of the compounds, thereby optimizing their pharmacological performance.

On days 10 and 30, animals were euthanized with carbon dioxide (carbon dioxide was introduced into the cage at a flow rate of 1.92 L per minute, with mice exposed to this carbon dioxide environment for a minimum of 5 min) after the end of the gavage treatment. The liver, spleen, kidneys, small intestine, seminal vesicle, testes, and epididymis were dissected and weighed. Trunk blood was collected, serum extracted by centrifugation at 4000× *g* for 15 min at 4 °C, and stored at −20 °C for hormone analysis. Small sections of liver and small intestine (including anterior, middle, and posterior segments) were frozen in liquid nitrogen and stored at −80 °C for later CYP3A4 enzyme and protein content analysis. All methods were in accordance with the AVMA Guidelines for the Euthanasia of Animals: 2020 Edition.

#### 2.2.1. Sperm Density and Motility

The cauda of epididymis was minced in physiological saline at 37 °C. The sperm suspension was diluted 1:10 and incubated for 5 min, and a drop was placed on a hemocytometer. Sperm in five squares were counted (n) to calculate density using the formula: sperm density (cells/mL) = n × 5 × 10 × 10^3^ × 10 [53]. Here, the term ‘sperm density’ refers to the concentration of sperm cells per milliliter in the prepared suspension. Based on WHO guidelines [54], sperm motility is classified into four categories: rapid progressive (a): sperm that move swiftly in a straight line, demonstrating strong forward motion; slow progressive (b): sperm that exhibit movement but at a slower pace, either moving forward gradually or in a less direct manner; non-progressive (c): sperm that are motile but do not move in a forward direction, exhibiting circular or random motion; and immotile (d): sperm that do not exhibit any movement. The motility rate was calculated as (a + b + c)/total sperm count [55].

#### 2.2.2. Sperm Abnormality Rate

Sperm morphology was analyzed by spreading a drop of sperm suspension on a clean slide, air-drying it, and fixing it with 95% alcohol for 10 min. The slide was then stained with 1–2% eosin, rinsed, dried, and covered with a cover glass. A total of 200 sperm were examined under the Evos FL Auto2 microscope (Thermo Fisher Scientific Inc., Waltham, MA, USA) to determine the percentage of abnormalities, following WHO guidelines [54]. The most common abnormalities, primarily affecting the head and tail, included banana-shaped, round-headed, double-headed, tail-folding, double-tailed, and hookless forms.

#### 2.2.3. LH and T Concentrations

Blood was collected from the retro-orbital plexus using a capillary tube and allowed to clot for 30 min at room temperature. After clotting, samples were centrifuged at 4000 rpm for 15 min to obtain serum, and the serum was stored at −20 °C in 1.5 mL centrifuge tubes for analysis of luteinizing hormone (LH) and testosterone (T) and hormone levels using ELISA kits (Good elisakit producers, Shanghai, China).

#### 2.2.4. CYP3A4 Contents in Small Intestine and Liver

Liver and small intestine sections (duodenum, jejunum, ileum) were snap-frozen in liquid nitrogen and stored at −80 °C. CYP3A4 enzyme content was quantified using ELISA kits with a 50 μL sample volume per well. The assay’s detection limit was below 1.0 pg/mL, with a range of 31.25–1000 pg/mL. Intra-assay and inter-assay CVs were <10% and <15%, respectively. The protein content in liver and small intestine tissues was measured using the Lowry method. Bovine serum albumin (BSA) was used to generate the standard curve. Tissue homogenates were prepared at a 1:9 (*w*/*v*) ratio in physiological saline and diluted 100-fold. A 50 μL aliquot of the diluted homogenate was added to a 96-well plate, followed by the addition of 250 μL of Lowry reagent. After incubating at 25 °C and 700 rpm for 10 min, 25 μL of Folin–phenol reagent was added, and the plate was incubated for another 30 min. The absorbance was measured at 500 nm, and the protein content was calculated based on the standard curve, with values multiplied by 1000 to account for the dilution factor.

### 2.3. Statistical Analysis

Statistical analysis was performed using SPSS 22.0. The Shapiro–Wilk test confirmed that the data were normally distributed (*p* > 0.05), and Levene’s test indicated homogeneity of variance across groups (*p* > 0.05). The effects of different treatments on organ weight (liver, kidney, spleen, small intestine, seminal vesicle, epididymis, and testis), sperm density, proportion of abnormal sperm, proportion of motile sperm, proportion of different type sperm, serum LH and T concentration, and liver and small intestine CYP3A4 content were examined by one-way ANOVA. The differences between groups were tested by LSD multiple comparison test. The results were presented as the mean ± standard error (mean ± SE), and statistical significance was defined as *p* < 0.05.

## 3. Results

### 3.1. Internal Organs of Male Mice After 10 d and 30 d

After 10 d, liver weight increased by 11.63%, 19.87%, 0.20%, 49.81%, and 34.84% in Q1, Q1 + K0.4, Q1 + K2, Q5, and Q5 + K0.4 compared to CK1, with significant increases in Q5 and Q5 + K0.4 (*p* < 0.05, Figure 1A). No significant differences in small intestine weight were found across groups (*F*_5,46_ = 1.574, *p* = 0.186, Figure 1B). Kidney weight was significantly lower in Q1 + K0.4 and Q5 compared to CK1 (*p* < 0.05), but no significant differences were observed in other groups (*F*_5,46_ = 2.115, *p* = 0.080, Figure 1C). Spleen weight varied significantly (*F*_5,46_ = 3.358, *p* = 0.011), increasing by 23.85%, 49.81%, 3.26%, 25.45%, and 28.53% in Q1, Q1 + K0.4, Q1 + K2, Q5, and Q5 + K0.4 compared to CK1, with significant increases in Q1 + K0.4 and Q5 + K0.4 (Figure 1D). Co-administration with ketoconazole did not significantly affect the internal organ weights compared to quinestrol alone except for a significant increase in small intestine weight in Q5 + K0.4 versus Q5 (Figure 1).

After 30 d, liver weight differed significantly among groups (*F*_5,46_ = 3.857, *p* = 0.005), decreasing by 16.60% and 3.02% in Q1 + K2 and Q5 + K0.4 and increasing by 9.74%, 1.05%, and 18.50% in Q1, Q1 + K0.4, and Q5 compared to CK2 (Figure 1E). Small intestine weights decreased by 8.56% in Q1 + K2 and by 8.83% in Q5 + K0.4 compared to CK2 as well as 10.4% in Q1 + K2 compared to Q1 and 16.2% in Q5 + K0.4 compared to Q5 (*p* < 0.05, Figure 1F). Spleen weight varied significantly (*F*_5,46_ = 1.887, *p* = 0.016), with Q5 showing significantly higher spleen weight compared to other groups (*p* < 0.05, Figure 1H). No significant differences were found between Q1 + K0.4 and Q1 in liver, small intestine, kidney, or spleen weights, but differences were noted between Q5 + K0.4 and Q5 (Figure 1).

### 3.2. Reproductive Organs and Sperm Density of Male Mice After 10 d and 30 d

After 10 d, testicular weight did not show significant differences among groups (*F*_5,46_ = 0.615, *p* = 0.689, Figure 2A). However, epididymal weight decreased significantly in the Q1 + K0.4, Q5, and Q5 + K0.4 groups by 5.65%, 21.97%, and 13.62% compared to CK1 (*F*_5,46_ = 6.379, *p* < 0.001, Figure 2B). Q1 + K0.4 showed an 11.03% lower epididymal weight compared to Q1. Seminal vesicle weight decreased significantly in Q1, Q1 + K2, Q5, and Q5 + K0.4 by 32.17%, 47.67%, 63.22%, and 56.70% compared to CK1 (*p* < 0.05), with no significant difference between CK1 and Q1 + K0.4 (*p* > 0.05, Figure 2C). Sperm density decreased significantly in Q1, Q1 + K0.4, Q1 + K2, Q5, and Q5 + K0.4 by 18.00%, 50.83%, 18.11%, 30.85%, and 33.94% compared to CK1 (*F*_5,46_ = 8.454, *p* < 0.001), with Q1 + K0.4 showing a 50.83% reduction compared to CK1 and 40.04% compared to Q1 (Figure 2D). Ketoconazole co-administration reduced sperm density more than quinestrol alone, but no significant changes were observed in reproductive organ weights compared to CK1. Q1 + K0.4 showed a significant decrease in sperm density compared to CK1 and Q1.

After 30 d, no significant differences were found in testis weight (*F*_5,46_ = 0.366, *p* = 0.869), epididymis weight (*F*_5,46_ = 0.825, *p* = 0.538), seminal vesicle weight (*F*_5,46_ = 1.998, *p* = 0.097), and sperm density (*F*_5,46_ = 0.416, *p* = 0.835) among the groups (Figure 2E–H). Seminal vesicle weight decreased by 11.18%, 19.76%, 26.48%, and 25.22% in Q1, Q1 + K0.4, Q5, and Q5 + K0.4 compared to CK2 (Figure 2G). Overall, Q1 + K0.4 significantly reduced sperm density at 10 days but had no effect on reproductive organ weights (Figure 2).

### 3.3. Sperm Abnormality Rate and Vitality of Male Mice After 10 d and 30 d

After 10 d, no significant differences in sperm abnormality rates were observed among the groups (*F*_5,46_ = 1.280, *p* = 0.289) except for Q1 + K0.4, which showed a significantly higher rate compared to CK1 and Q1 (*p* < 0.05, Figure 3A). Sperm motility significantly decreased across groups, with the most pronounced reduction in Q1 + K0.4. (*F*_5,46_ = 19.383, *p* < 0.001, Figure 3B). Significant differences were found in the proportions of rapid progressive (*F*_5,46_ = 11.588, *p* < 0.001), slow progressive (*F*_5,46_ = 19.767, *p* < 0.001), non-progressive (*F*_5,46_ = 6.984, *p* < 0.001), and immotile sperm (*F*_5,46_ = 19.383, *p* < 0.001) among the groups, with the greatest reduction in rapid progressive sperm and the largest increase in immotile sperm in Q1 + K0.4 compared to CK1 (Figure 3C). Overall, after 10 days, sperm abnormality rates and motility were significantly affected in the Q1 + K0.4 group, with ketoconazole co-administration further reducing sperm motility.

After 30 d, there was no significant difference in sperm motility among the groups (*F*_5,45_ = 2.289, *p* = 0.062), although Q5 and Q5 + K0.4 was significantly reduced by 5.04% and 3.75% compared to CK2 (*p* < 0.05, Figure 3D). Significant differences were observed in the proportions of rapid progressive sperm (*F*_5,45_ = 4.362, *p* = 0.003), slow progressive sperm (*F_5,45_* = 4.735, *p* = 0.001), non-progressive sperm (*F*_5,45_ = 2.760, *p* = 0.029), and immotile sperm (*F*_5,45_ = 2.458, *p* = 0.047) among groups (Figure 3E), with reductions in progressive and slow progressive sperm and an increase in immotile sperm in Q1, Q1 + K0.4, Q1 + K2, Q5, and Q5 + K0.4 compared to CK2 (Figure 3E).

### 3.4. LH and T Concentrations in Serum of Male Mice After 10 d and 30 d

After 10 d, no significant difference in serum LH among the groups was found (*F*_5,46_ = 2.348, *p* = 0.056). Serum LH concentration was reduced by 58.18%, 17.56%, 61.19%, and 74.36% in Q1, Q1 + K0.4, Q1 + K2, Q5, and Q5 + K0.4 compared to CK1 (Figure 4A). Significant differences were observed in serum T concentration among groups (*F*_5,46_ = 2.482, *p* = 0.045). with reductions of 73.25%, 72.08%, and 75.09% in the Q1, Q1 + K0.4, and Q5 groups, respectively, compared to CK1 (*p* < 0.05, Figure 4B).

After 30 d, LH levels decreased by 14.51%, 17.5%, 74.56%, 19.77%, and 75.27% in Q1, Q1 + K0.4, Q1 + K2, Q5, and Q5 + K0.4 groups compared to CK2, with significant increases observed in Q1 + K2, Q5, and Q5 + K0.4 (*p* < 0.05). The co-administration of ketoconazole with quinestrol led to a more pronounced reduction in LH levels compared to quinestrol alone (Figure 4C). Significant differences in T levels were observed among the groups (*F*_5,46_ = 3.775, *p* = 0.006), with T levels decreasing by 56.83%, 52.53%, 40.59%, and 15.64% in Q1, Q1 + K2, Q5, and Q5 + K0.4 compared to CK2 (Figure 4D).

### 3.5. CYP3A4 Content in Small Intestine of Male Mice After 10 d and 30 d

After 10 d, significant differences in CYP3A4 content per gram of small intestine tissue between groups (*F*_5,46_ = 2.758, *p* = 0.029, Figure 5A). However, no significant differences were found in CYP3A4 content per gram of small intestine protein (*F*_5,46_ = 1.668, *p* = 0.161) or in total CYP3A4 content in the small intestine (*F*_5,46_ = 2.088, *p* = 0.084, Figure 5B,C). The Q1 + K2 and Q5 + K0.4 groups exhibited lower CYP3A4 levels in all measurements compared to the quinestrol-alone group and CK1.

After 30 d, significant differences in both CYP3A4 content per gram of small intestine tissue (*F*_5,46_ = 3.336, *p* = 0.012) and total CYP3A4 content in the small intestine (*F*_5,46_ = 4.500, *p* = 0.002) were detected among the groups. The co-administration of ketoconazole reduced CYP3A4 compared to quinestrol alone (Figure 5D,F). Although the difference in CYP3A4 content per gram of small intestine protein was not statistically significant (*F*_5,46_ = 2.109, *p* = 0.082), a similar decreasing trend was observed with ketoconazole co-treatment (Figure 5E).

### 3.6. CYP3A4 Contents in Liver of Male Mice After 10 d and 30 d

After 10 d, CYP3A4 content per gram live tissue was significantly different among groups (*F*_5,46_ = 2.961, *p* = 0.021, Figure 6A). CYP3A4 content per gram of liver protein was significantly increased in Q1, Q1 + K0.4, Q1 + K2, Q5, and Q5 + K0.4 compared to CK1 by 33.93%, 50.94%, 33.56%, 54.26%, and 41.66% (*F*_5,46_ = 3.720, *p* = 0.008, Figure 6B). Total CYP3A4 content in live tissue also significantly increased in these groups compared to CK1 by 16.37%, 20.93%, 11.48%, 50.32%, and 53.31% (*F*_5,46_ = 6.222, *p* < 0.001, Figure 6C).

After 30 d, significant differences in CYP3A4 content per gram of live tissue among the groups were found (*F*_5,46_ = 7.748, *p* < 0.001). However, the combination treatment groups showed no significant difference compared to quinestrol alone (*p* > 0.05) (Figure 4D). No significant difference was observed in CYP3A4 content per gram of liver protein (*F*_5,46_ = 0.956, *p* = 0.457, Figure 6E). A significant difference in total CYP3A4 content in the liver was found (*F*_5,46_ = 6.12, *p* = 0.039), with the combination treatment groups having lower total CYP3A4 content in the liver compared to quinestrol alone (Figure 6F).

## 4. Discussion

The study demonstrates that quinestrol, both alone and in combination with ketoconazole, effectively suppressed male mice’s reproductive parameters in a dose-dependent manner. Significant reductions in seminal vesicle weight, sperm density, sperm motility, and rapid sperm rate, along with increases in slow and immotile sperm rates, were observed across most treatment groups. Previous studies have consistently shown that quinestrol effectively suppresses male reproductive parameters across various rodent species, reducing testis and epididymis size, sperm count, and fertility. Dosages ranging from 0.33 mg/kg to 100 mg/kg over periods from 5 to 90 days have demonstrated significant impacts, including complete sterility in some cases [22,27,29,48,56,57,58]. These findings collectively illustrate the potent reproductive suppression achieved with quinestrol across different rodent species despite variations in dosage and administration duration.

In this study, quinestrol treatment resulted in significant liver and spleen hypertrophy in male rats, with the least enlargement observed in the Q1 + K2 group after 10 days. By 30 days, the liver weight decreased in the Q1 + K2 and Q5 + K0.4 groups while increasing in others. These findings align with prior studies, such as Wang et al., which demonstrated significant liver enlargement in female mice at quinestrol doses above 0.04 mg/kg, without impacting the small intestine or kidneys [55]. Similarly, a 28.64% increase in liver weight was noted after bi-weekly administration of 0.2 mg/kg quinestrol for 14 doses [21], and a 35.54% increase was observed with 4 mg/kg quinestrol in female Meriones unguiculatus over three consecutive days at 25-day intervals [20]. Our research indicates that the co-administration of ketoconazole attenuates quinestrol-induced liver enlargement, especially at higher doses. Importantly, this combination did not significantly affect small intestine or kidney weights, suggesting that ketoconazole selectively mitigates hepatotoxic effects without broadly impacting other organs, thus presenting a potential strategy to reduce adverse effects. The significant increases in liver weight observed in the Q5 and Q5 + K0.4 groups after 10 days likely stem from the metabolic burden imposed by quinestrol. As a known CYP3A4 inhibitor, ketoconazole may reduce quinestrol clearance, resulting in increased hepatic exposure and subsequent hypertrophy, a dose-dependent effect consistent with previous findings. Conversely, the lack of significant changes in small intestine, kidney, and spleen weights indicates that these organs have a limited role in quinestrol metabolism. Overall, our findings emphasize the liver’s sensitivity to quinestrol treatment, particularly at higher doses, underscoring its pivotal role in processing this compound, especially in conjunction with metabolic inhibitors like ketoconazole.

Variations in quinestrol’s effects on reproductive organs across studies may be due to differences in dosage, treatment duration, species, and experimental conditions. In our study, significant reductions in epididymal weight, seminal vesicle weight, and sperm density were observed after 10 days, particularly with quinestrol combined with ketoconazole, suggesting a potential synergistic effect. However, by 30 days, these effects were less pronounced, indicating possible adaptation or recovery. These results align with studies showing that high doses of quinestrol lead to significant reductions in reproductive organ weights and sperm density [22,59]. In contrast, other studies report notable decreases in testicular and epididymal weights primarily at higher doses (100 mg/kg), with lower doses (5 mg/kg, 10 mg/kg, 50 mg/kg) having minimal impact [30]. Species-specific responses or interactions with other compounds like ketoconazole may explain these discrepancies. After 10 days, the Q1 + K0.4 group showed the most severe declines in sperm density and motility, with increased sperm abnormalities. This suggests that quinestrol and ketoconazole impair sperm quality, potentially due to damage to epididymal cells and decreased fructose secretion from seminal vesicles [29,60,61]. After 30 days, while overall sperm motility was not significantly different, variations in motility types persisted, indicating a partial recovery. In summary, quinestrol treatment, especially combined with ketoconazole, significantly impacts reproductive organ weights and sperm quality. After 30 days, though no major differences in organ weights or sperm density were noted, some effects on sperm motility and quality remained, suggesting that long-term effects on sperm parameters may not directly correlate with organ weight changes. Further research is needed to explore these interactions and their long-term implications for reproductive health.

In this study, we observed significant reductions in testosterone (T) levels in Q1, Q1 + K0.4, and Q5 after 10 days of treatment, even though luteinizing hormone (LH) concentrations remained unchanged. This finding aligns with previous research indicating that higher doses of quinestrol can lower T levels without affecting LH levels [55]. Contrarily, some studies have reported increased T levels following short-term quinestrol administration in male mice [62], while others noted no significant impact on T at lower doses in *Lasiopodomys brandtii* [27]. Such discrepancies emphasize the influence of dosage, treatment duration, and species on quinestrol’s effects. After 30 days, LH levels were consistent across all groups, suggesting a potential recovery from quinestrol’s effects over time. Nevertheless, the reduction in both LH and T levels is often associated with reproductive organ atrophy [30], which negatively impacts sperm quantity and quality. Our study highlights that reduced reproductive organ weights directly correlate with compromised sperm production and maturation, a situation exacerbated by lower hormone levels. Although organ weights may eventually recover, the persistence of hormonal imbalances and impaired sperm quality underscores the need for the continuous monitoring of reproductive health post-treatment. Furthermore, significant alterations in LH and T levels were noted following 10 and 30 days of treatment with quinestrol and ketoconazole. These hormonal changes are critical to understanding the reproductive effects observed. LH and T play pivotal roles in male reproductive health by regulating spermatogenesis and maintaining reproductive organ function. A significant reduction in LH, as observed in our combined treatment groups (Q1 + K2, Q5 + K0.4), suggests disrupted hypothalamic–pituitary–gonadal (HPG) axis feedback, likely resulting in diminished spermatogenesis and potentially compromising sperm quality. Similarly, reduced T levels could adversely affect sperm production and male reproductive physiology. These findings, including observed reductions in sperm density and reproductive organ weights, suggest that hormonal imbalances induced by the treatments may impair reproductive health.

Over 10 days, quinestrol significantly increased CYP3A4 levels in the liver, while combination treatments with ketoconazole (Q1 + K2, Q5 + K0.4) reduced CYP3A4 content in the small intestine. After 30 days, this reduction in intestinal CYP3A4 persisted. Quinestrol alone markedly elevated liver CYP3A4, both in the total content and per gram of protein, particularly after 10 days. However, co-administration with ketoconazole reduced liver CYP3A4 levels, especially after 30 days, indicating ketoconazole’s inhibitory effect on hepatic CYP3A4 activity. These findings align with previous findings, where quinestrol treatment alone increased hepatic CYP3A4 and CYP1A2 levels in *Meriones unguiculatus* after five days of quinestrol treatment [20]. The biological mechanism behind this involves ketoconazole’s competitive inhibition of CYP3A4, leading to an accumulation of quinestrol and a more sustained impact on the reproductive system. These findings offer valuable insights into optimizing male contraceptive strategies by leveraging enzyme inhibition to maintain efficacy while minimizing dosage. Further studies are needed to assess the long-term impact of this combination therapy on reproductive health.

After 10 days of treatment, sperm density, motility, morphology, and serum hormone levels were significantly reduced, but these effects diminished after 30 days. The reduction in sperm density in the Q1 + K0.4 group suggests a synergistic effect of ketoconazole, likely due to its role as a CYP3A4 inhibitor, which increases quinestrol’s bioavailability by slowing its metabolism [20]. The co-administration of quinestrol and ketoconazole may allow for effective spermatogenesis suppression at lower doses, offering an optimized male contraceptive approach. Significant reductions in serum LH, especially in the Q1 + K2 and Q5 + K0.4 groups after 10 days, indicate the suppression of the HPG axis feedback mechanism [20]. However, this effect weakened after 30 days, possibly due to physiological adaptation or altered drug clearance. Lower CYP3A4 levels, particularly in the small intestine, suggest that ketoconazole modulates quinestrol’s metabolism, influencing its effectiveness and side effects. Decreased CYP3A4 was linked to reductions in epididymal and seminal vesicle weights, sperm density, and motility alongside increased sperm abnormalities. The decline in testosterone levels further underscores CYP3A4’s role in reproductive regulation. From a practical standpoint, the quinestrol–ketoconazole combination shows promise in wildlife population control. The combination of quinestrol with 0.4 mg/kg ketoconazole produced marked reductions in reproductive organ weight and sperm density after 10 days, without affecting internal organ weights. By using a CYP3A4 inhibitor, effective reproductive suppression can be achieved with lower quinestrol doses, reducing side effects.

Future research should focus on the long-term effects of quinestrol and ketoconazole on reproductive health, especially regarding hormonal balance, sperm quality, and organ recovery. Variations in response across species, dosages, and treatment durations highlight the need for comparative studies. Further investigation into how ketoconazole modulates quinestrol’s effects through CYP3A4 inhibition is also crucial for optimizing the dosage and reducing side effects. Studies should explore the recovery of reproductive function after treatment cessation to assess the potential for long-term or reversible reproductive impairment. A limitation of this study is the lack of long-term toxicity evaluations. While increased liver and spleen size was observed in treatment groups, no other overt toxic effects were detected. However, the possibility of chronic toxicity and long-term hormonal imbalances should be explored, given ketoconazole’s known hepatotoxicity. The significant reductions in serum testosterone and LH levels could lead to broader physiological consequences, such as reduced bone density or metabolic disruption. Future studies should assess the long-term safety of the quinestrol–ketoconazole combination, focusing on potential liver toxicity, hormonal changes, and irreversible reproductive damage.

## 5. Conclusions

Based on the results, the combination of quinestrol and ketoconazole, particularly the Q1 + K0.4 treatment, exhibited the most pronounced effects in reducing reproductive capacity. After 10 days, this combination significantly reduced epididymal weight (11.03% lower compared to quinestrol alone) and seminal vesicle weight and caused the largest decrease in sperm density (50.83% reduction compared to the control). The Q1 + K0.4 group also showed marked reductions in serum testosterone levels, corresponding with decreased reproductive organ weights and compromised sperm quality. Although differences in reproductive organ weights and sperm density among the groups lessened after 30 days, suggesting a potential reversibility of quinestrol’s effects on reproductive parameters, the Q1 + K0.4 treatment continued to trend toward reduced seminal vesicle weight and lower testosterone levels. Furthermore, the lack of significant increases in CYP3A4 content in intestinal and liver tissues with this combination suggests strong metabolic inhibition, which may contribute to its pronounced effects on reproductive parameters. In conclusion, the Q1 + K0.4 combination demonstrated the most substantial impact among those tested, as evidenced by significant reductions in key reproductive parameters, indicating that ketoconazole greatly enhances quinestrol’s effects on reproductive capacity.

## Figures and Tables

**Figure 1 animals-14-03240-f001:**
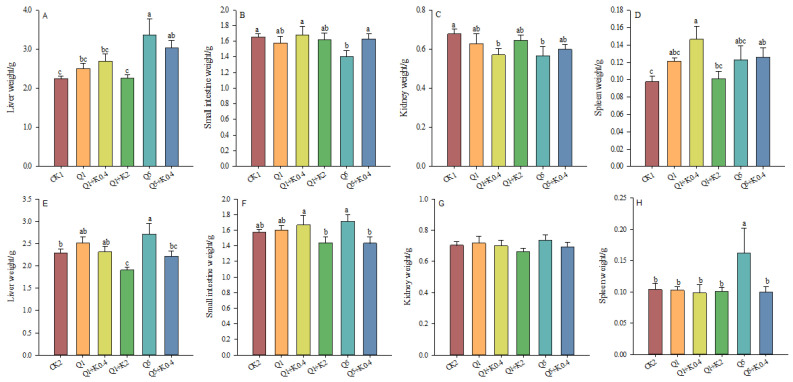
Effects of combined ketoconazole and quinestrol treatment on internal organs after 10 days(**A**–**D**) and 30 days (**E**–**H**). Note: Male mice in groups CK1 and CK2 were gavaged with sunflower seed oil, while groups Q1, Q1 + K0.4, Q1 + K2, Q5, and Q5 + K0.4 received sunflower seed oil + 1.0 mg/kg quinestrol, 0.4 mg/kg ketoconazole + 1.0 mg/kg quinestrol, 2.0 mg/kg ketoconazole + 1.0 mg/kg quinestrol, sunflower seed oil + 5.0 mg/kg quinestrol, and 0.4 mg/kg ketoconazole + 5.0 mg/kg quinestrol, respectively. Different lowercase letters indicate significant differences between groups, LSD, *p* < 0.05; values were mean ± SE.

**Figure 2 animals-14-03240-f002:**
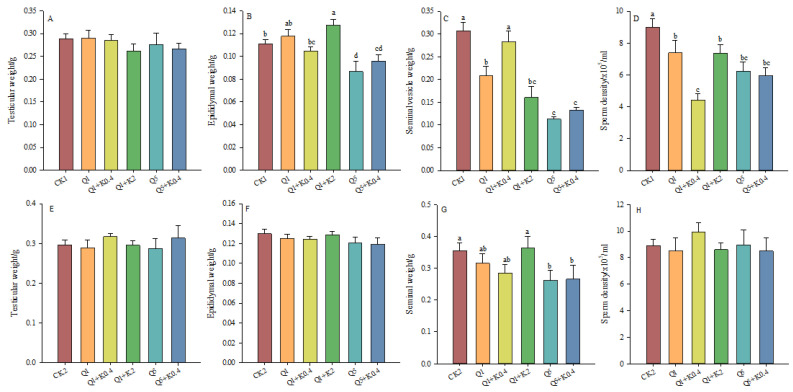
Effects of combined ketoconazole and quinestrol treatment on reproductive organs and sperm density after 10 days (**A**–**D**)and 30 days(**E**–**H**). Note: Male mice in groups CK1 and CK2 were gavaged with sunflower seed oil, while groups Q1, Q1 + K0.4, Q1 + K2, Q5, and Q5 + K0.4 received sunflower seed oil + 1.0 mg/kg quinestrol, 0.4 mg/kg ketoconazole + 1.0 mg/kg quinestrol, 2.0 mg/kg ketoconazole + 1.0 mg/kg quinestrol, sunflower seed oil + 5.0 mg/kg quinestrol, and 0.4 mg/kg ketoconazole + 5.0 mg/kg quinestrol, respectively. Different lowercase letters indicate significant differences between groups, LSD, *p* < 0.05; values were mean ± SE.

**Figure 3 animals-14-03240-f003:**
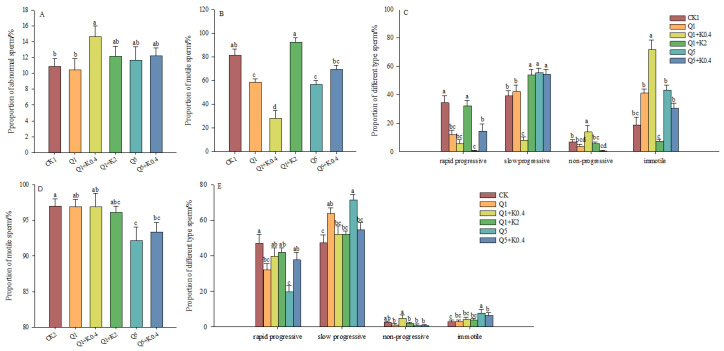
Effects of combined ketoconazole and quinestrol treatment on sperm abnormality rate (**A**), sperm motility (**B**), and sperm motility characteristics (**C**) after 10 days and sperm motility (**D**) and sperm motility characteristics (**E**) after 30 days. Note: Male mice in groups CK1 and CK2 were gavaged with sunflower seed oil, while groups Q1, Q1 + K0.4, Q1 + K2, Q5, and Q5 + K0.4 received sunflower seed oil + 1.0 mg/kg quinestrol, 0.4 mg/kg ketoconazole + 1.0 mg/kg quinestrol, 2.0 mg/kg ketoconazole + 1.0 mg/kg quinestrol, sunflower seed oil + 5.0 mg/kg quinestrol, and 0.4 mg/kg ketoconazole + 5.0 mg/kg quinestrol, respectively. Different lowercase letters indicate significant differences between groups, LSD, *p* < 0.05; values were mean ± SE.

**Figure 4 animals-14-03240-f004:**
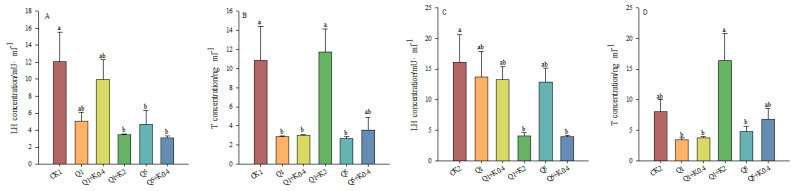
Effects of combined ketoconazole and quinestrol treatment on serum LH (**A**) and T (**B**) concentrations after 10 days and serum LH (**C**) and T (**D**) concentrations after 30 days. Note: Male mice in groups CK1 and CK2 were gavaged with sunflower seed oil, while groups Q1, Q1 + K0.4, Q1 + K2, Q5, and Q5 + K0.4 received sunflower seed oil + 1.0 mg/kg quinestrol, 0.4 mg/kg ketoconazole + 1.0 mg/kg quinestrol, 2.0 mg/kg ketoconazole + 1.0 mg/kg quinestrol, sunflower seed oil + 5.0 mg/kg quinestrol, and 0.4 mg/kg ketoconazole + 5.0 mg/kg quinestrol, respectively. Different lowercase letters indicate significant differences between groups, LSD, *p* < 0.05; values were mean ± SE.

**Figure 5 animals-14-03240-f005:**
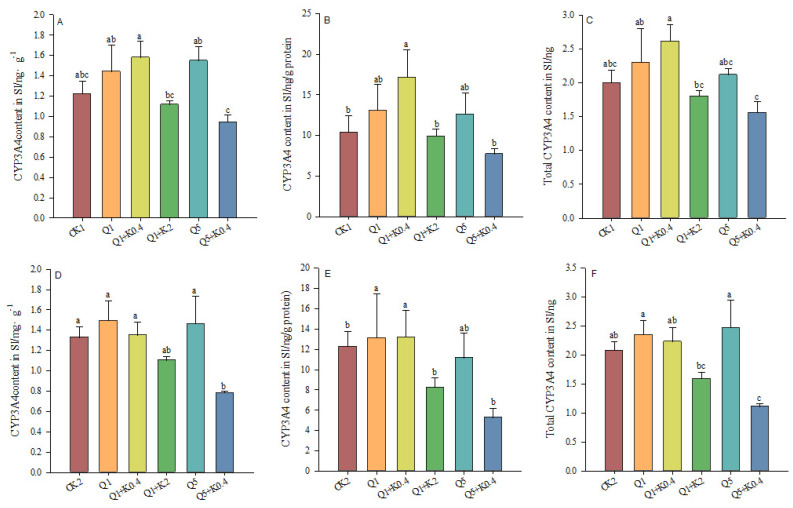
Effects of combined ketoconazole and quinestrol treatment on CYP3A4 content (ng/g) (**A**), CYP3A4 content (ng/g protein) (**B**), and total CYP3A4 content (ng) (**C**) in small intestine after 10 days and CYP3A4 content (ng/g) (**D**), CYP3A4 content (ng/g protein) (**E**), and total CYP3A4 content (ng) (**F**) in small intestine after 30 days. Note: Male mice in groups CK1 and CK2 were gavaged with sunflower seed oil, while groups Q1, Q1 + K0.4, Q1 + K2, Q5, and Q5 + K0.4 received sunflower seed oil + 1.0 mg/kg quinestrol, 0.4 mg/kg ketoconazole + 1.0 mg/kg quinestrol, 2.0 mg/kg ketoconazole + 1.0 mg/kg quinestrol, sunflower seed oil + 5.0 mg/kg quinestrol, and 0.4 mg/kg ketoconazole + 5.0 mg/kg quinestrol, respectively. Different lowercase letters indicate significant differences between groups, LSD, *p* < 0.05; values were mean ± SE.

**Figure 6 animals-14-03240-f006:**
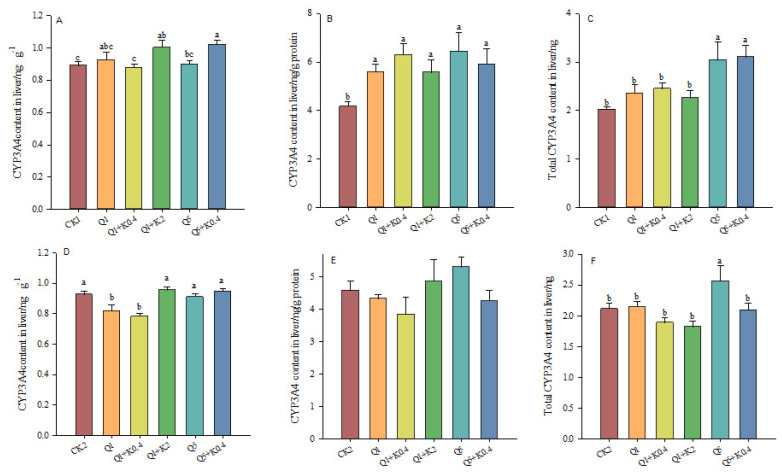
Effects of combined ketoconazole and quinestrol treatment on CYP3A4 content in liver (ng/g) (**A**), CYP3A4 content in liver (ng/g protein) (**B**), and total CYP3A4 content in liver (ng) (**C**) after 10 days and CYP3A4 content in liver (ng/g) (**D**), CYP3A4 content in liver (ng/g protein) (**E**), and total CYP3A4 content in liver (ng) (**F**) after 30 days. Note: Male mice in groups CK1 and CK2 were gavaged with sunflower seed oil, while groups Q1, Q1 + K0.4, Q1 + K2, Q5, and Q5 + K0.4 received sunflower seed oil + 1.0 mg/kg quinestrol, 0.4 mg/kg ketoconazole + 1.0 mg/kg quinestrol, 2.0 mg/kg ketoconazole + 1.0 mg/kg quinestrol, sunflower seed oil + 5.0 mg/kg quinestrol, and 0.4 mg/kg ketoconazole + 5.0 mg/kg quinestrol, respectively. Different lowercase letters indicate significant differences between groups, LSD, *p* < 0.05; values were mean ± SE.

**Table 1 animals-14-03240-t001:** Combination of different dosages and experimental time.

Total Days	Group	Components 1	Components 2	Number	Gavage Time
10 d	CK1	Sunflower seed oil	Sunflower seed oil	15	3 d
	Q1	Quinestrol 1.0 mg/kg	Sunflower seed oil	7	3 d
	Q1 + K0.4	Quinestrol 1.0 mg/kg	Ketoconazole 0.4 mg/kg	7	3 d
	Q1 + K2	Quinestrol 1.0 mg/kg	Ketoconazole 2.0 mg/kg	8	3 d
	Q5	Quinestrol 5.0 mg/kg	Sunflower seed oil	7	3 d
	Q5 + K0.4	Quinestrol 5.0 mg/kg	Ketoconazole 0.4 mg/kg	8	3 d
30 d	CK2	Sunflower seed oil	Sunflower seed oil	15	3 d
	Q1	Quinestrol 1.0 mg/kg	Sunflower seed oil	7	3 d
	Q1 + K0.4	Quinestrol 1.0 mg/kg	Ketoconazole 0.4 mg/kg	7	3 d
	Q1 + K2	Quinestrol 1.0 mg/kg	Ketoconazole 2.0 mg/kg	8	3 d
	Q5	Quinestrol 5.0 mg/kg	Sunflower seed oil	7	3 d
	Q5 + K0.4	Quinestrol 5.0 mg/kg	Ketoconazole 0.4 mg/kg	8	3 d

## Data Availability

The data presented in this study are available on request from the corresponding author.

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
