# Peer review of "The Effect of Ketoconazole and Quinestrol Combination on Reproductive Physiology in Male Mice"

_animals, 2024, doi:10.3390/ani14223240_

Round 1

Reviewer 1 Report

Comments and Suggestions for Authors

This study presents a novel combination of quinestrol and ketoconazole, offering a potential new strategy for fertility control in males. The concept of using a CYP3A4 inhibitor to enhance the efficacy of quinestrol is promising. The results demonstrate that the Q1+K0.4 regimen significantly reduces key reproductive parameters while effectively inhibiting CYP3A4. These findings suggest practical applications for lower doses of quinestrol.

There are some concerns related to the study and manuscript which should be considered while revising the manuscript:

1.   The study primarily addresses short-term effects (10 and 30 days) without adequate long-term follow-up. This raises concerns about the sustainability of the anti-fertility effects and the potential recovery after treatment cessation.

2.   While increase in liver and spleen size with quinestrol is noted, the study does not provide sufficient detail on potential long-term toxicity and other side effects of the tested drug combination

3.   While the use of male mice is appropriate, the applicability of these findings to human or rodent populations remains unclear. The oral administration of the drugs presents a major drawback when evaluated for rodent management.

4.   Study does not discuss the correlation between hormonal imbalances and reproductive health.

5.   The potential side effects of the ketoconazole and quinestrol combination are not explored.

6.   There is no mention of the concentrations of hormones and proteins evaluated in the abstract.

7.   The criteria followed to distribute rats into different groups need clarification.

8.   It should be mentioned in the materials and methods section that the ketoconazole and quinestrol solutions were sonicated separately. Additionally, clarify why the doses of the two were administered separately.

9.   Provide detailed information on how blood samples were collected and serum separated. Note that centrifugation at 4000 rpm for 15 minutes obtains plasma, not serum. Clarify whether LH and testosterone were tested in serum or plasma.

10.           What was the exact concentration of eosin used to stain the epididymal smear for sperm abnormalities?

11.           The methods section lacks mention of rotational sperm motility, which is otherwise reported in the results. Clarify what is meant by "rotational sperms," and provide data on progressive and non-progressive sperm motility.

12.           Why was there no observed effect of treatment on the prostate gland?

13.           There is no mention of the method used for estimating total soluble proteins in methods section.

14.           Spelling mistakes and typographical errors should be corrected.

15.           What was the reason for the low sperm density observed in the CK1 group?

16.           There is considerable variability in the results, with no significant effect of different treatments on organ weights, except for an increase in liver weight in a few groups.

17.           The percent decrease in different parameters is not revealed in any table. The tables are not reader-friendly; it would be better to use figures to compare the groups. Figures should also illustrate effects related to reversibility in different groups. But table and figures should not repeat the data

18.               Consider adding more and latest relevant references from across the globe. like   Integrative Zoology, 19(1), 108-126.

Comments on the Quality of English Language

Some spelling mistakes and typographical errors should be corrected.

Reviewer 2 Report

Comments and Suggestions for Authors

1.      In the title all the words are starting with caps, the authors can change quinestrol also into capital Q

2.      In the authors affiliation # is used but not specified the role of #

3.      The keyword can be little more specific , the authors can add  CYP3A4, the word fertility control is repeated twice in the keywords section, you can delete one

4.      The authors have mentioned that the animals were maintained in a plastic cage, are they BPA free plastic cages? What type of water bottles used? BPA free?

5.      In materials and methods “2.2 Experimental treatment” line no.126, the author has cited “15” reference , what does it mean? Have you followed their methodology?

6.      In the methodology section “The treatment group” is not well defined and more confusive for the readers, hence the authors can make it as a table format or other easier way for the readers to understand.

7.      Since the authors have obtained permission for more than 100 animals, they could have left few treated animals without sacrificing for mating and breeding to see the fertility success rate and reproductive behaviour

8.      If n=52 and 6 groups, how many animals per group were maintained?

9.      How did you administer the dosage? Since odour of ketoconazole not pleasant, did they consumed the full dose? Or force fed them?

10.   Why author is expecting weight loss in the liver, spleen intestine & kidney?

11.    In the results section most of the reproductive parameters were showing no significant changes

 “After 10d, testicular weight did not show significant differences among groups”

“After 30d, no significant differences were found between the six groups for testis Weight”

“After 10d, sperm abnormality rates were not significantly”

“After 30d, there was no significant difference in sperm motility”

“After 10d, there was no significant difference in serum LH”

12.   But in the discussion author says “The study demonstrates that quinestrol, both alone and combination with ketoconazole, effectively suppressed male mice’ reproductive parameters”

13.   The author may explain what type of interaction between quinestrol and ketoconazole chemically happens to bring out these changes in reproductive parameters

14.   What is the possible mechanism by which these chemicals act on endocrine system and metabolism

Reviewer 3 Report

Comments and Suggestions for Authors

Concerning study tittled " The Effect of Ketoconazole and quinestrol Combination on Re- 2 productive Physiology in Male Mice ", I found that there are no citation for this research.  I suggest major revision. 

The manuscript study few parameters of reproductive physiology including weight of reproductive organs, sperm analysis, hormonal dosage and completely miss histological analysis of testis or epidedymis. Histological analysis of reproductive organ such as testis is essential in the study of male reproductive function. Therefore, I kindly request author to add the histological analysis of testis or epedidymis to the manuscript.

Authors investigate the potential inhibitor/suppressive effect of Ketoconazole and quinestrol on male reproductive function in mice. The idea of the research work is very interesting, however many points should be revised before considering the manuscript for future publication.

The Abstract and introduction parts are clear and well aiming.

Material and methods: The manuscript study few parameters of reproductive physiology including weight of reproductive organs, sperm analysis, hormonal dosage and completely miss histological analysis of testis or epidedymis. Histological analysis of reproductive organ such as testis is essential in the study of male reproductive function. Therefore, I kindly request author to add the histological analysis of testis or epedidymis to the manuscript.

The dose chosen for Quinestrol and for Ketoconazole should be justified with reference.

The experimental design is complicated and should be clarified to the reader by adding a design Flowchart of the different experimental sets.

Line 143 insert amount of used material (carbon dioxide) for anesthesia

Insert company, country and catalog code of material (microscope, elisa kits…)

Sperm analysis:

Text which part of epididymis used for analysis? Cauda, caput

Please add reference for the formula of sperm density calculation  

Reference should be added to the section describing sperm morphology analysis.

Results section needed adjustments related to sentence reformulation in a less cumbersome manner.

Discussion

Author should justify and explain the choice of liver and intestine for weight and CYP3A4 content measurement, rather than reproductive organ such as testis or epididymis

Conclusion summarizes appropriately the obtained findings. However, authors should suggest further related researches being based on the raised assumptions from obtained findings (perspective).

Round 2

Reviewer 1 Report

Comments and Suggestions for Authors

The authors have answered all the queries. 

Reviewer 3 Report

Comments and Suggestions for Authors

The authors of the manuscript have answered all my questions and suggestions for changes and have incorporated them into the manuscript. I recommend accepting the manuscript as it is.
